# Let's Agree to Disagree on Operative versus Nonoperative (LADON) treatment for proximal humerus fractures: Study protocol for an international multicenter prospective cohort study

Ruben J. Hoepelman[1,2]*, Yassine Ochen[1,3], Frank J. P. Beeres[4], Herman Frima[5], Christoph Sommer[6], Christian Michelitsch[6], Reto Babst[4,7], Isabelle R. Buenter[4], Detlef van der Velde[3], Egerbert-Jan M. M. Verleisdonk[2], Rolf H. H. Groenwold[8,9], Roderick M. Houwert[1], Mark van Heijl[1,2]

1 Department of Trauma Surgery, UMC Utrecht, Utrecht, The Netherlands, 2 Department of Trauma Surgery, Diakonessenhuis Utrecht, Utrecht, The Netherlands, 3 Department of Trauma Surgery St. Antonius Hospital Nieuwegein, Nieuwegein, The Netherlands, 4 Department of Trauma Surgery, Cantonal Hospital of Lucerne, Luzerne, Switzerland, 5 Department of Trauma Surgery, Noordwest Ziekenhuisgroep Alkmaar, Alkmaar, The Netherlands, 6 Department of Trauma Surgery, Kantonsspital Graubünden, Chur, Switzerland, 7 Department of Health Science and Medicine, University of Lucerne, Lucerne, Switzerland, 8 Department of Clinical Epidemiology, Leiden University Medical Center, Leiden, The Netherlands, 9 Department of Biomedical Data Sciences, Leiden University Medical Center, Leiden, The Netherlands

* rjhoepelman@gmail.com

**Funding:** This study was funded by DePuy Synthes for the completion of the study. No author

## Abstract

### Background

The proximal humerus fracture is a common injury, but the optimal management is much debated. The decision for operative or nonoperative treatment is strongly influenced by patient specific factors, regional and cultural differences and the preference of the patient and treating surgeon. The aim of this study is to compare operative and nonoperative treatment of proximal humerus fractures for those patients for whom there is disagreement about optimal management.

### Methods and analysis

This protocol describes an international multicenter prospective cohort study, in which all patients of 18 years and older presenting within three weeks after injury with a radiographically diagnosed displaced proximal humerus fracture can be included. Based on patient characteristics and radiographic images several clinical experts advise on the preferred treatment option. In case of disagreement among the experts, the patient can be included in the study. The actual treatment that will be delivered is at the discretion of the treating physician. The primary outcome is the QuickDash score at 12 months. Propensity score matching will be used to control for potential confounding of the relation between treatment modality and QuickDash scores.

personally received specific awards. Grant number DPS-TCMF-2019-033). The organization has not been involved in the drafting of this protocol or review of this manuscript. They will not be involved in study design, data collection, analysis or the decision to publish the final manuscript. URL: https://www.jnjmedicaldevices.com/en-EMEA/companies/depuy-synthes.

**Competing interests:** The authors have declared that no competing interests exist.

## Discussion

The LADON study is an international multicenter prospective cohort study with a relatively new methodological study design. This study is a "natural experiment" meaning patients receive standard local treatment and surgeons perform standard local procedures, therefore high participation rates of patients and surgeons are expected. Patients are only included after expert panel evaluation, when there is proven disagreement between experts, which makes this a unique study design. Through this inclusion process, we create two comparable groups whom received different treatments and where expert disagree about the already initiated treatment. Since we are zooming in on this particular patient group, confounding will be largely mitigated. Internationally the treatment of proximal humerus fractures are still much debated and differs much per country and hospital. This observational study with a natural experiment design will create insight into which treatment modality is to be preferred for patients in whom there is disagreement about the optimal treatment strategy.

## Trial registration

Registered in Netherlands trial register NL9357 and Swiss trial register CH 2020–00961; https://clinicaltrials.gov/.

## Introduction

The proximal humerus fracture is a common injury, accounting for 5.7% of all fractures and the third most common fracture type in the elderly population [1,2]. The optimal management of acute proximal humerus fractures, i.e. operative or nonoperative treatment, is much debated [3–5]. Several meta-analyses have been inconclusive whether operative treatment of proximal humerus fractures is superior to nonoperative treatment [3–5].

The Proximal Fracture of the Humerus Evaluation by Randomization (PROFHER) study, a multicenter randomized controlled trial including 250 patients, is to date the largest study that compared outcomes following operative and nonoperative treatment of proximal humerus fractures. The PROFHER trial evaluated the Oxford Shoulder Score, the Short-Form 12, complications, subsequent therapies, and mortality. The study found no statistically significant or clinically relevant differences in outcomes between the operative and nonoperative treatment groups [6]. However, the validity and generalizability of these findings have been called into question [7,8]. The inclusion of patients in the PROFHER trial was not consecutive. In total, 1250 patients were screened, of which 563 patients were found eligible for inclusion. However, only 250 patients eventually consented to take part in the study. Furthermore, the 109 patients who received operative treatment were treated by 66 different surgeons in 30 different hospitals. The low number of cases per surgeon could have influenced the results of this study. What is more, since the inclusion rate is so low, generalizing results from such a selective group is not straightforward [9].

In practice, patient-specific factors can have a large impact on the decision for operative or nonoperative management, which leads to challenges during surgical trials. Furthermore, both patients and surgeons can have a strong preference for a certain treatment, which forms an obstacle for randomization in surgical trials [10]. The patient populations encountered in daily clinical practice often differ from the highly selected patient populations enrolled in randomized controlled trials [11].

In contrast, observational studies tend to have much larger sample sizes than randomized trials and thus provide an opportunity to investigate a variety of patient populations [12,13]. Observational studies could complement results of randomized trials, provided incomparability of patients who receive different treatment modalities (i.e., confounding) is adequately controlled for. A natural experiment, based on e.g. practice variation, could be considered, for example when recommendations for operative or nonoperative management are largely influenced by training of the treating surgeons [14]. For instance, a natural experiment could be set up by comparing two countries with a different preference for operative or nonoperative treatment of proximal humerus fractures, such as Switzerland and the Netherlands, respectively [15,16]. The aim of this multicenter international prospective cohort study is to compare outcome after management of proximal humerus fractures between Switzerland and the Netherlands by evaluating outcomes of patient populations in daily clinical practice, where disagreement exists about operative or nonoperative treatment.

## Methods and analysis

The LADON study will be an international multicenter prospective cohort study with the following participating centers from the Netherlands (University Medical Centre Utrecht (UMCU), Diakonessenhuis (DIAK), Sint Antonius Ziekenhuis (SAZ), where the predominant preference is for nonoperative treatment and from Switzerland (Cantonal Hospital of Lucerne (LUKS) and Kantonsspital Graubünden (KSGR)) with a predominant preference for operative treatment. This study protocol was written in adherence to the Standard Protocol Items: Recommendations for Interventional Trials (SPRIT) guideline [17].

### Participants selection

**Inclusion criteria.**   All patients (>18 years) presenting with an acute displaced proximal humerus fracture involving minimally the surgical neck, including isolated greater tuberosity fractures are eligible for inclusion. Patient enrollment started the first of July 2020.

**Exclusion criteria.**   Exclusion criteria include; open fracture, pre-existing co-morbidities which preclude operative treatment, pathological fractures, associated dislocation of injured shoulder joint, associated ipsilateral upper extremity fractures, concomitant soft tissue injury or neurovascular injuries requiring operative treatment, delayed presentation (> three weeks after injury), treatment for re-fractures, cognitive impairment, non-Dutch, non-German, or non-English speaking patients, patients not resident in the hospitals area and unavailable for follow-up.

### Intervention

The decision for operative or nonoperative treatment is left to the treating orthopedic trauma surgeon at the participating hospitals. The actual treatment will be initiated prior to inclusion in the LADON study.

**Nonoperative treatment.**   Nonoperative treatment consists of sling immobilization for three to six weeks for comfort, adequate pain management and guided physiotherapy according to local protocol.

**Operative treatment.**   The operative treatment consists of open reduction and internal plate fixation (ORIF), minimal invasive plate osteosynthesis (MIPO), intramedullary nailing, or arthroplasty of the glenohumeral joint. The decision for the type of operative treatment is left to the treating physician. Peri-operative management including anesthesia, antibiotics and thromboembolism prophylaxis will follow the national guidelines and local protocol.

## Expert panel progress

After informed consent, pseudonymized data sets including radiographs, AO fracture classification, and baseline patient characteristics during first presentation will be collected. The individual patient data sets will be made available on a secure online platform. All relevant data to reach a "clinical" decision will be made available, including basic clinical information, radiographs, key images of CT-scans if available and radiology reports. The data will be presented to an expert panel of each country, blinded to the already initiated treatment. The expert panels will consist of an equal distribution of representatives from both countries, three orthopedic trauma surgeons from the participating Dutch centers and three orthopedic trauma surgeons from participating Swiss centers. Members of each panel will independently decide on the preferred management, operative or nonoperative treatment, for each individual case. Patients can be included in the study if the majority of experts in one country, i.e., minimally two out of three, disagree with the received treatment in the other country. For example, patients are eligible if they would have received operative treatment based on the expert opinions in e.g. Switzerland, while in fact they received nonoperative treatment in the Netherlands, and vice versa. Patients will be excluded when the majority of the panel in one country agree on the received treatment in the other country. This will lead to a group of patients for whom there is disagreement on the optimal management, operative or nonoperative treatment. Both operative and nonoperative treated patients will be send to the expert panels to ensure blinding. A flow chart of the patient recruitment is shown in Fig 1. For practical efficiency we chose to perform the expert panel evaluation prior to inclusion rather than include all eligible patients and perform an expert panel evaluation afterwards.

In an ideal observational study all patient characteristics are measured and adjusted for. Instead of this utopia we try to identify patient subgroups for whom their profile is likely similar. We assume the patient profile is a continuum, such that on one end of the continuum all patients will receive conservative treatment, whereas on the other end of the continuum all patients will receive operative treatment. However, different hospitals (or countries) may have different preference for operative or conservative treatment, which provides a contrast that forms the basis for an observational study of these treatments. By zooming in on a patient with a similar profile, confounding by patient characteristics will, to a large extent, be mitigated. One way of identifying patients with a similar profile is to identify those patients for whom experts disagree about the appropriate treatment. The more we zoom in, the better we adjust for confounding, however, the less precise estimates will be due to fewer patients being included in the study (Fig 2).

## Outcome measures

**Primary objectives.** Primary outcome is assessed using the QuickDash score at 12 months following treatment. The QuickDASH is a patient-reported outcome instrument developed to measure upper extremity disability and symptoms, resulting in a score ranging from no disability (0) to most severe disability (100) [18].

**Secondary objectives.** Secondary functional outcomes include the QuickDash score at six weeks, the Subjective shoulder value (SSV), EuroQol five dimensional questionnaire (EQ-5D), Numerical Rating Scale (NRS) Pain score, return to sporting activity and return to work activity at 6 weeks and 12 months. The SSV is a subjective value for shoulder function expressed as a percentage of an un injured shoulder, which would score 100% [19]. The EQ-5D is a standardized questionnaire for generic health status measurements to asses quality of life [20].

Other secondary outcomes will include complications, revision surgery, implant removal and related complications. Complications will include cases of non-union, mal-union,

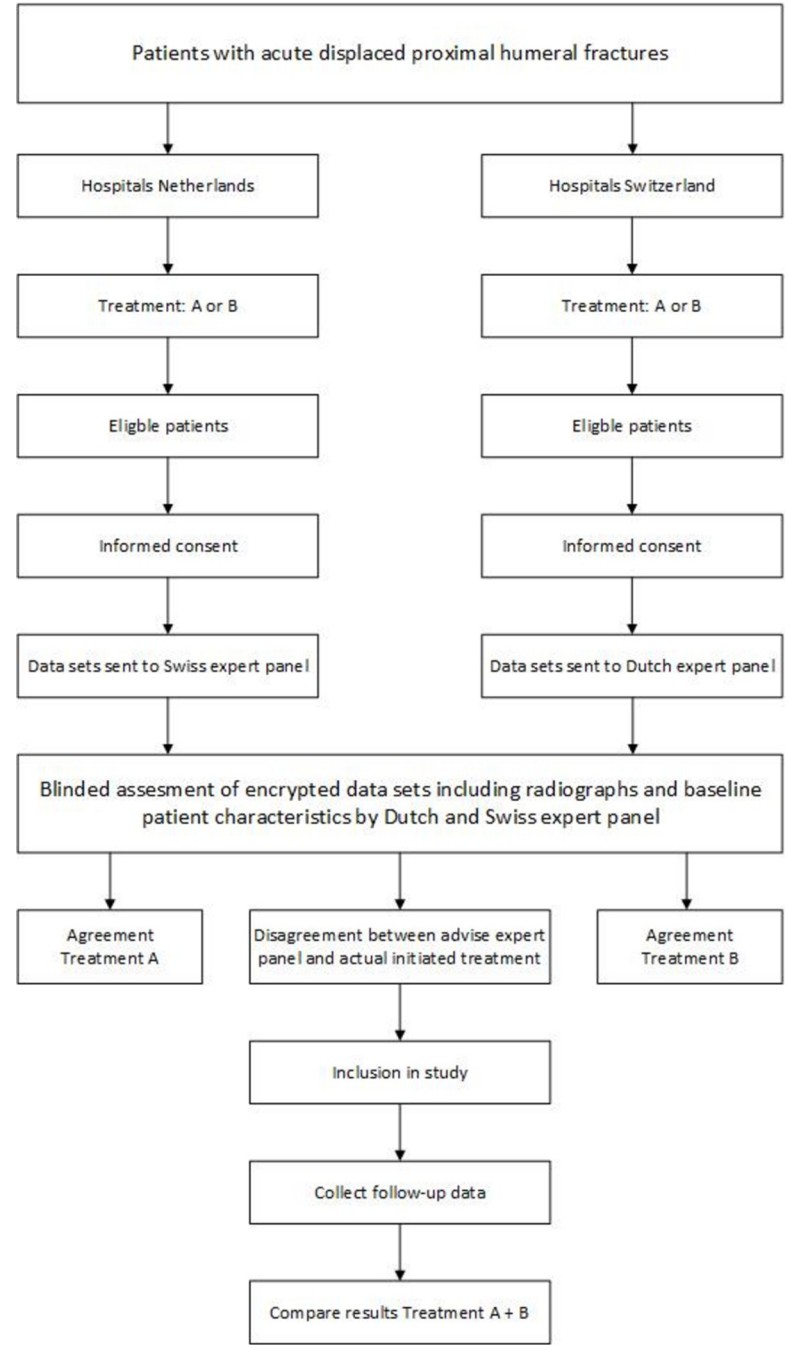

**Fig 1. Flow chart of the patient recruitment.**

superficial infection and fracture related infection, implant failure, confirmed deep venous thrombosis, myocardial infarction or stroke. Non-unions is defined as a minimum of six months with persistent pain and no signs of fracture healing on radiographic images. Mal-union defined as fracture union in an incorrect anatomical position on a radiograph or CT scan resulting in pain. Superficial infection is defined as redness, swelling, and/or purulent discharge from the wound that could be treated with oral antibiotics or wound incision. Fracture

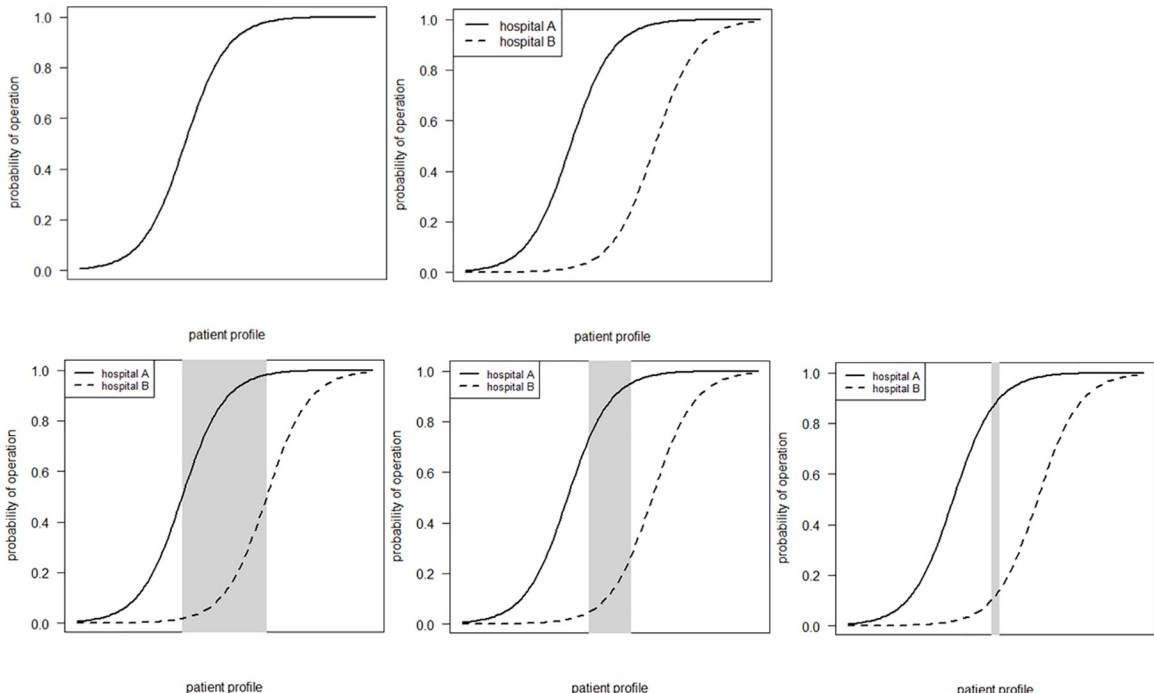

**Fig 2. Patient profile continuum and confounding.** Patient profile is a continuum, on the one end all patients receive nonoperative treatment and on the other end all patients receive operative treatment, this can differ per country. By zooming in on patients with a similar profile, confounding will be mitigated to a large extent.

related infection is defined according to Metsemakers et al [21]. Implant failure is defined as loss of reduction, implant dislocation, implant breakage or breakage of screws. Revision surgery is defined as the need for secondary surgical treatment other than implant removal.

## Data collection and patients follow-up

All patients will be reviewed at the standard six weeks and 12 months outpatient clinic visits after treatment. Standardized patient-reported outcome questionnaires will be collected during the outpatient visit, by performing telephone interviews or using a secure online questionnaire platform: Castor Electronic Data Capture (EDC) system. Electronic medical records will be reviewed to collect baseline characteristics regarding history of previous injuries of the affected shoulder, age, sex, BMI, ASA sore, trauma date, trauma mechanism, time from injury to treatment, current profession, and physical demands. Data collection will be performed by reviewing electronic medical records, operative reports, radiology reports and patient interviews by independent research fellows. All data will be stored in a research folder that can only be accessed by the principal investigator and independent research fellows. Data will be pseudonymized and a file to decode these data will be stored in a research folder only accessible by the principal investigator and independent research fellows.

## Sample size considerations

As indicated above, the primary outcomes are defined as the difference in patient-reported functional outcome as measured by the QuickDASH, measured at 12 months after the injury. The developers of the QuickDASH report that a difference of eight points represents a clinically relevant difference for discriminating between improved and stable patients [22]. A

previous observational cohort study by Frima et a.l [23], performed by our research group, found patients who had MIPO surgery had a QuickDASH standard deviation (SD) of 14. Sample size calculations indicate that a study with a power of 80% and two-sided statistical significance level (alpha) of 0.05 would require 2 x 50 participants. Accounting for 20% loss to follow-up and an additional 40% loss after propensity score matching (used to account for confounding, see Statistical Analysis), the total sample size is 220 participants.

**Feasibility.** The planned recruitment period to reach the required sample size is based on retrospective data on proximal humerus fractures in the participating centers. The study started simultaneously in the participating centers on July 1st 2020. To reach the required sample size the recruitment period will be 24 months and an additional 12 months will be needed to collect follow-up data on all patients (Fig 3). Based on observational data from the participating hospitals, approximately 550 patients with acute displaced proximal humerus fractures are treated each year in both the Netherlands and Switzerland. In the Netherlands up to 90% of the patients with fractures around the shoulder are treated conservatively and in Switzerland up to 40% are treated operatively [15,16]. We performed a preliminary expert panel evaluation, which resulted in an overall disagreement rate of approximately 40%. The proposed recruitment period of 24 months would, after accounting for a 40% disagreement rate and loss of ineligible patients, result in 220 patients for analysis.

## Statistical analysis

Descriptive results will be presented as mean values with standard deviations and range (SD, range), median values with interquartile range (IQR) or absolute numbers and percentages (%). To control for potential confounding, we will conduct propensity score matching. The propensity score will be estimated using binary logistic regression analysis, with operative or nonoperative treatment as the dependent variable and age, sex, BMI, ASA score, AO classification as covariates in the model. The primary analysis will be performed within the dataset of propensity score matched patients. Normal distribution will be confirmed using the Shapiro–Wilk test. Correlation between continuous variables will be measured using the Pearson correlation coefficient. Missing data will not be imputed. Differences between study groups will be analysed using the independent sample t-test, Mann–Whitney U test, or the Chi-square test. A two-tailed p-value less than 0.05 will be considered significant.

## Discussion

The LADON study is an international multicenter prospective cohort study with a relatively new methodological study design. The aim of the study is to compare nonoperative and operative treatment of proximal humerus fractures. It is a "natural experiment", meaning patients receive standard local treatment and surgeons perform standard local procedures. Therefore high participation rates of patients and surgeons are expected. Furthermore, patients are only included after expert panel evaluation when there is proven disagreement between experts, which makes this a unique study design. Through this inclusion process, we create two comparable groups whom received different treatments and where expert disagree about the already initiated treatment. Since we are zooming in on this particular patient group, confounding will be mitigated. In addition, propensity score matching will be performed. Despite several trials and meta-analysis, the treatment of proximal humerus fractures are still much debated internationally and differs much per country and hospital. This observational study with a natural experiment design will create insight into which treatment modality is to be preferred for patients in whom there is disagreement about the optimal treatment strategy.

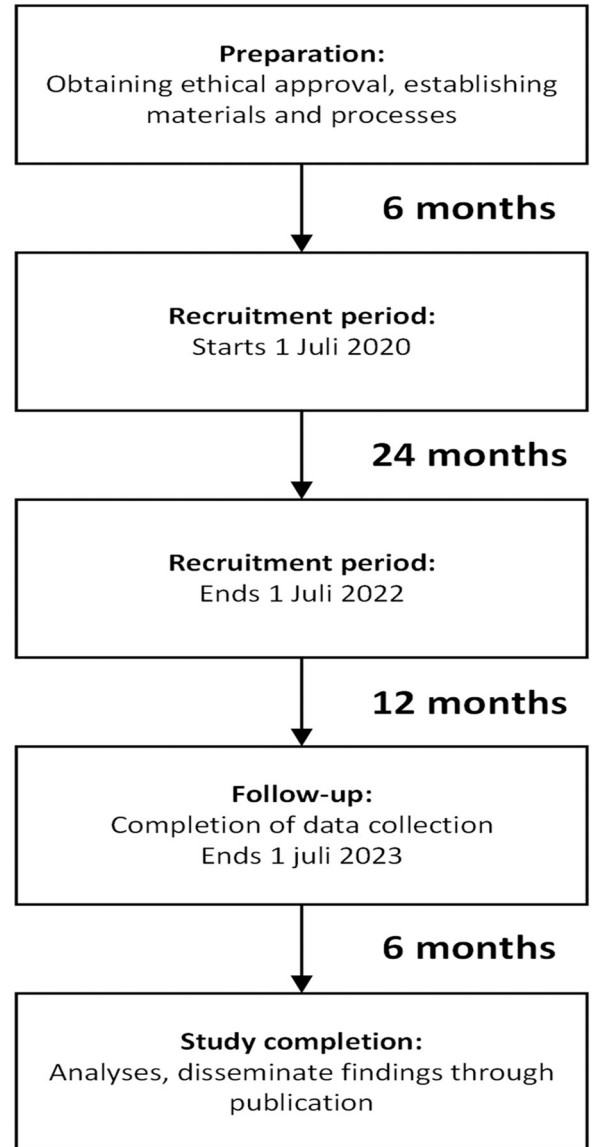

**Fig 3. Timeline LADON study.**

## Trial status

This is protocol version 2, dating from 28[th] February 2020. Recruitment began 1[st] of July 2020. Inclusion of patients is expected to be completed at July 2022. So far 102 patients from the Netherlands have been included, which are at the moment only conservatively treated patients. Switzerland has included 80 patients so far, which are exclusively operatively treated patients so far.

## Supporting information

**S1 Checklist. SPIRIT 2013 checklist: Recommended items to address in a clinical trial protocol and related documents**[*]**.**
(DOC)

**S1 File. LADON data management statement.**
(DOCX)

**S2 File. Supplementary material WHO data set.**
(DOCX)

## Acknowledgments

This work was part of the activities of the Natural Experiments Study Group (www.next-studygroup.org).

## Ethical approval and consent to participate

The Medical Ethics Committee Utrecht (METC-U) and the Medical Research Ethics Committees United (MEC-U) confirmed that the Medical Research Involving Human Subject Act (WMO) does not apply to the study (METC-protocol number 20-169/C). The Ethikkommission Nordwest- und Zentralschweiz (EKNZ) also approved the study (proposal number 2020–00961). Patients will be fully informed of the purpose and procedures of the study by the treating surgeon or a designated researcher, and signed informed consent will be obtained in agreement with the General Data Protection Regulation. Study results will be submitted for peer review publication.

## Author Contributions

**Conceptualization:** Yassine Ochen, Frank J. P. Beeres, Rolf H. H. Groenwold, Roderick M. Houwert, Mark van Heijl.

**Investigation:** Ruben J. Hoepelman.

**Methodology:** Ruben J. Hoepelman, Yassine Ochen, Frank J. P. Beeres, Rolf H. H. Groenwold, Mark van Heijl.

**Visualization:** Ruben J. Hoepelman.

**Writing – original draft:** Ruben J. Hoepelman, Yassine Ochen, Rolf H. H. Groenwold, Roderick M. Houwert, Mark van Heijl.

**Writing – review & editing:** Ruben J. Hoepelman, Yassine Ochen, Frank J. P. Beeres, Herman Frima, Christoph Sommer, Christian Michelitsch, Reto Babst, Isabelle R. Buenter, Detlef van der Velde, Egerbert-Jan M. M. Verleisdonk, Rolf H. H. Groenwold, Roderick M. Houwert, Mark van Heijl.

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
