## [Decision Letter · Decision Letter 0]

11 Feb 2022

Let's Agree to Disagree on Operative versus Nonoperative (LADON) treatment for proximal humerus fractures: study protocol for an international multicenter prospective cohort study

PONE-D-22-00924

Dear Dr. Hoepelman,

We’re pleased to inform you that your manuscript has been judged scientifically suitable for publication and will be formally accepted for publication once it meets all outstanding technical requirements.

Kind regards,

Zsolt J. Balogh, MD, PhD, FRACS

Academic Editor

PLOS ONE

Additional Editor Comments (optional):

Reviewers' comments:

Reviewer's Responses to Questions

**Comments to the Author**

1. Does the manuscript provide a valid rationale for the proposed study, with clearly identified and justified research questions?

Reviewer #1: Yes

Reviewer #2: Yes

2. Is the protocol technically sound and planned in a manner that will lead to a meaningful outcome and allow testing the stated hypotheses?

Reviewer #1: Yes

Reviewer #2: Partly

3. Is the methodology feasible and described in sufficient detail to allow the work to be replicable?

Reviewer #1: Yes

Reviewer #2: Yes

4. Have the authors described where all data underlying the findings will be made available when the study is complete?

Reviewer #1: Yes

Reviewer #2: Yes

5. Is the manuscript presented in an intelligible fashion and written in standard English?

Reviewer #1: Yes

Reviewer #2: Yes

6. Review Comments to the Author

You may also provide optional suggestions and comments to authors that they might find helpful in planning their study.

Reviewer #1: This is a well described study protocol. I wonder how it will turn out in practice. That however is not the point of the paper which is to describe the a study protocol.

Reviewer #2: I think this is an interesting study with a new methodology. As the primary outcome is measured at one year, I am not sure if the study can show the advantage of one method over the other as recovery will plateau off earlier.

Also the rehabilitation protocol will be heterogeneous so one method may work at one center and may not at another.

7. PLOS authors have the option to publish the peer review history of their article (what does this mean?). If published, this will include your full peer review and any attached files.

Reviewer #1: **Yes: **Minoo Patel

Reviewer #2: No

---

## [Editor Report · Acceptance letter]

17 Feb 2022

PONE-D-22-00924 

Let’s Agree to Disagree on Operative versus Nonoperative (LADON) treatment for proximal humerus fractures: study protocol for an international multicenter prospective cohort study 

Dear Dr. Hoepelman:

I'm pleased to inform you that your manuscript has been deemed suitable for publication in PLOS ONE. Congratulations! Your manuscript is now with our production department. 

Kind regards, 

on behalf of

Dr. Zsolt J. Balogh 

Academic Editor

PLOS ONE